# Passive Sonar Target Identification Using Multiple-Measurement Sparse Bayesian Learning

**DOI:** 10.3390/s22218511

**Published:** 2022-11-04

**Authors:** Myoungin Shin, Wooyoung Hong, Keunhwa Lee, Youngmin Choo

**Affiliations:** Department of Ocean Systems Engineering, Sejong University, Seoul 05006, Korea

**Keywords:** passive sonar system, frequency detection, beamforming tracking, sparse Bayesian learning

## Abstract

Accurate estimation of the frequency component is an important issue to identify and track marine objects (e.g., surface ship, submarine, etc.). In general, a passive sonar system consists of a sensor array, and each sensor receives data that have common information of the target signal. In this paper, we consider multiple-measurement sparse Bayesian learning (MM-SBL), which reconstructs sparse solutions in a linear system using Bayesian frameworks, to detect the common frequency components received by each sensor. In addition, the direction of arrival estimation was performed on each detected common frequency component using the MM-SBL based on beamforming. The azimuth for each common frequency component was confirmed in the frequency-azimuth plot, through which we identified the target. In addition, we perform target tracking using the target detection results along time, which are derived from the sum of the signal spectrum at the azimuth angle. The performance of the MM-SBL and the conventional target detection method based on energy detection were compared using in-situ data measured near the Korean peninsula, where MM-SBL displays superior detection performance and high-resolution results.

## 1. Introduction

The passive sonar system receives the underwater acoustic signals composed of narrowband and broadband components from marine objects, such as surface ships, submarines, and fishing boats. Target detection using acoustic measurements is important for identifying and tracking underwater target signals [1,2,3]. Techniques, such as energy detection (ED) [4,5,6], constant false alarm rate (CFAR) [7,8,9], machine learning (ML) [10,11,12,13,14,15,16,17,18], and compressive sensing (CS) [19,20,21,22,23,24,25,26], have been proposed for efficient target detection.

The traditional target detection method is the ED, which estimates the energy according to each azimuth and frequency component through broadband processing. Subsequently, a detection threshold technique was applied. A representative ED method is the conventional energy detection (CED). The basic principle of CED is the spatial coherence of the frequency components contained in the target signal, which is in a specific azimuth. Because the frequency components of the target signal are spatially aligned, the energy estimation for the azimuth is strengthened and the detectability of the target signals is increased [4,5]. However, owing to its limited azimuth resolution, CED generates wide contact traces and has the limitation of poor performance in a real acoustic environment with multiple signals [5]. To overcome this limitation, sub-band energy detection (SED), which sums the energy of the peaks and valleys in the azimuth spectrum for each frequency bin, has been proposed [6]. The SED is advantageous for broadband target signal detection, but its application is limited because the detection performance is degraded for a target emitting a narrowband target signal [6].

CFAR improves target detection performance by adjusting the threshold according to the measured background. Several CFAR schemes exist, such as cell-averaging CFAR, and ordered sort CFAR [7,8]. CFAR schemes do not require prior information about the environment or noise and have the same false alarm probability regardless of the noise variance for the given detector decision threshold [7]. However, the detection performance using CFAR schemes is poor in an underwater environment where clutter and noise generated by terrain or obstacles are mixed [9].

Machine learning has been applied to beamforming, classification, depth estimation, and target detection [10,11,12,13,14,15,16,17,18] in underwater acoustics. Even though ML-based schemes have achieved great scientific results, their application is limited because sufficient training data and hyperparameters that users need to tune and optimize are required [18].

Compressive sensing reconstructs sparse signals represented by a linear combination of few meaningful components using limited observations [19,20]. CS has high-resolution performance results, and it has been applied to various underwater acoustic fields, such as beamforming, detection, and line spectral estimation [21,22,23,24,25,26]. However, there are several limitations, such as the computational burden, depending on the size of the linear system model and regularization parameter setup that controls the sparsity of solution. To overcome these limitations, sparse Bayesian learning (SBL) has been applied to underwater acoustic fields. SBL was first proposed for classification and regression in machine learning [27]. Recently, in underwater acoustics, SBL has been applied to beamforming [28,29,30], localization [31,32], mode extraction [33], and line spectral estimation [34]. In the SBL framework, the noise and source power variances are automatically obtained through an iterative optimization process. Moreover, owing to the sparsity of the signals, the solution of the linear system using SBL is advantageous in suppressing noise and obtaining high-resolution performance. In addition, the multiple-measurement SBL (MM-SBL) has been proposed to improve performance using multiple measurements, which increasing the robustness of the SBL against noise by using the commonality of the source signals [28,34].

Traditional target identification estimates the target’s azimuth by increasing the signal-to-noise ratio (SNR) of the target signal through beamforming and then identifies the target through frequency analysis at a specific azimuth angle. In this paper, we simultaneously considered multiple measurements received by the passive sonar systems. Unlike the conventional target identification method, we first detected the common frequency components of signals using MM-SBL. Then, we performed direction of arrival (DOA) estimation for each common frequency component using MM-SBL based beamforming. After that, on the frequency-azimuth plot, we confirmed the azimuth of the signal for each common frequency component and identified the target. In addition, the path of the target can be tracked by arranging its DOA estimation results, which are indicated by each detected common frequency component along time.

The paper is organized as follows. In Section 2, the conventional target detection method is presented. Section 3 introduces the system model for the frequency analysis and DOA estimation with the theoretical background of the MM-SBL. Section 4 introduces the proposed target identification method using MM-SBL. Section 5 provides an in-situ data description and application results of the proposed identification method. In Section 6, a brief discussion is given. Finally, Section 7 summarizes the present study.

## 2. Conventional Target Detection Method

In a passive sonar system, received signals generally have four types of signals: tonal signals (generated by the operation of the machinery of the ship), propeller noise (generated by the cavitation which is produced by the propeller rotating), hydrodynamic noise (generated by friction between the ship and the fluid), and ambient noise [1]. A passive sonar system generally suppresses other types of signals except the tonal signal, by applying an analog filter (or low-pass filter) to the received data [34]. Therefore, in this paper, we detect marine objects using the filtered data dominated by the tonal signals.

The conventional target detection technique in passive sonar systems is the ED method. Beams are formed using the sensor array to detect the target. A signal in a specific azimuth is strengthened through beamforming, and a fast Fourier transform (FFT) is performed on each beam. Based on these results, the frequency components of the target were displayed according to the azimuth angle in the frequency-azimuth (FRAZ) plot.

Figure 1a shows the FRAZ plot for a single time scan of the in-situ data using the ED method. The data used in Figure 1 were measured in an environment where the experimental ship and several fishing boats existed and detection using the ED method showed unclear detection results. Figure 1b shows the results of applying the local maximum and detection thresholds to Figure 1a to reduce the noise signal and improve detection performance. We obtained the local maximum values for the frequency and steering angle axes and retained components with amplitudes greater than the threshold as detection signals. However, this result shows that the detection results vary depending on the threshold value designated by human opinion. Many false alarms occur (for example, at approximately 0.4 and 0.8 of normalized frequency).

Therefore, in this paper, to overcome these limitations in the conventional target detection method, we performed target detection using MM-SBL. MM-SBL first detects the common frequency components of the received signal from the passive sonar system and then evaluates the azimuth angle of each common frequency component. Furthermore, we identified the target by displaying the detection results on the FRAZ plot. MM-SBL attenuates the noise signal and derives high resolution detection results.

## 3. Target Identification and Tracking Using MM-SBL

In this section, we introduce target identification and the tracking method using the MM-SBL scheme. A passive sonar system consists of a sensor array and each sensor receives data that has common information of the target signals. Therefore, we performed a frequency analysis and DOA estimation using SBL with multiple measurements. MM-SBL suppresses noise and provides high-resolution results by finding sparse common components in multiple measurement data.

The traditional target identification method generates multiple beams and increases SNR through beamforming. Subsequently, a frequency analysis is performed on the beam containing the target, and the target is identified according to the frequency characteristics. In this paper, unlike the traditional target identification method, we first found the common frequency components of the signals received by the array through the MM-SBL based frequency analysis and then estimated the azimuth of each common frequency component through the MM-SBL based beamforming.

Figure 2 presents a flowchart of target identification and tracking using MM-SBL. We considered input data to be the filtered time-domain signals measured by the sensor array of the passive sonar system. As previously mentioned, because passive sonar systems consist of multiple sensors, we considered temporal (multiple time snapshots) and spatial (multiple sensors) multiple measurements in frequency analysis. An SBL using multiple measurements improves the detection performance and robustness of the SBL against noise by emphasizing the commonality of the signals contained in the data. Therefore, frequency analysis using the MM-SBL leads to consistent frequency estimates by diminishing the random noise components. We detect the frequency components of the target signals in the data through a frequency analysis using MM-SBL. This study aimed to detect continuously occurring tonal frequency components in the underwater targets. Therefore, to detect the frequency components of the target signals more accurately, we performed a frequency analysis three times, where each frequency analysis used one second shifted signal over time. Then, we select components that exactly match in each frequency analysis result as the common frequency components. Although, this method could lead to the loss of some of the tonal frequency components of the target signals; however, it shows that it has sufficient detection performance (shown in Section 5).

Next, in order to find the azimuth of each detected common frequency component, we performed DOA estimation using narrowband beamforming based on MM-SBL, which considers temporal (multiple-time snapshots) multiple measurements. DOA estimation using the MM-SBL suppresses noise and has high-resolution performance owing to the sparsity of the target signals. On the FRAZ plot, we confirmed the azimuth of the target with each common frequency component and verified the characteristics of the frequency components configured for each azimuth angle. Based on these results, we could identify targets for each azimuth angle. In addition, the path of the target can be tracked using the target detection results along time, derived from the sum of the signal spectrum at an azimuth angle in the FRAZ plot.

## 4. System Model & Theoretical Background of MM-SBL

This section presents system models for frequency analysis and DOA estimation and describes the theoretical background of the MM-SBL.

### 4.1. System Model

As shown in Figure 3, frequency analysis is performed using the filtered discrete time series data as input and DOA estimation is performed at a specific frequency (narrowband beamforming) using the frequency domain data converted from the time-domain signals by FFT after being received by the sensor array. That is, frequency analysis and the DOA estimation were performed using inputs in different domains.

For the frequency analysis problem, y=[y1, y2,…,yMf]T∈ℝMf×1 is an Mf-sized real-valued filtered discrete time series vector; x∈ℂMf×1 is an unknown vector relevant to the frequency component amplitude, and it has Nf elements that determines the frequency resolution. In addition, n∈ℝMf×1 is the noise. A=[a(f1),a(f2),…,a(fNf)]∈ℂMf×Nf is a transformation matrix where fn=(n−1)/Nf with n∈[1,…,Nf],
(1)a(fn)=[1,ej2πfn·1,…,ej2πfn·(Mf−1)]T

For the DOA estimation problem, y=[y1,y2,…,yMd]T∈ℂMa×1 is a complex-valued measurement vector and ym is the FFT coefficient corresponding to the frequency of f at the mth sensor, where m∈[1,…,Ma] and Ma is the number of sensors. A=[a(θ1),a(θ2),…,a(θNa)]∈ℂMa×Na is a transformation matrix that contains the steering vector a(θn) for DOAs as the columns, θn=−90+((n−1)×180°)/Na with n∈[1,…,Na],
(2)a(θn)=[1,ej2π1·dλsin(θn),…,ej2π(Ma−1)·dλsin(θn)]T
where d is the gap between the sensors and λ is the wavelength. x∈ℂNa×1 is the unknown source amplitude, and n∈ℂMa×1 is the noise.

For both the frequency analysis and the DOA estimation problems, the linear system model relates the measured data y to the unknown source amplitude x as
(3)y=Ax+n
where A and n are the transformation matrix and noise, respectively [29,34]. For simplicity, we define $M_{f}=M_{a}=M$ and $N_{f}=N_{a}=N$.

Here, we considered multiple-measurement system models for the frequency analysis and DOA estimation. The passive sonar system comprises sensors that continuously measure signals of interest. Therefore, we can use multiple measurements with multiple sensors and multiple-time snapshots for the frequency analysis and use multiple measurements with multiple-time snapshots for the DOA estimation. The multiple-measurement system model of Equation (3) is givens as follows:(4)Y=AX+N
where $Y=\left[y_{1},y_{2},\ldots ,y_{L}\right]$ is the M×L multiple-measurement matrix, $X=\left[x_{1},x_{2},\ldots ,x_{L}\right]$ is the N×L unknown matrix, $N=\left[n_{1},n_{2},\ldots ,n_{L}\right]$ is the M×L noise matrix, and L is the number of multiple measurement vectors.

### 4.2. Multiple-Measurement Sparse Bayesian Learning

In the SBL framework, the unknowns X and noise N are treated as random matrices, which are assumed to follow a zero-mean complex Gaussian distribution (for the frequency analysis, we consider the zero-mean Gaussian distribution for the SBL (refer to [27])).

MM-SBL solves the linear system of Equation (4) using the given measurement Y by finding $\hat{X}$ which maximizes the following probability:(5)$$\hat{X}=\underset{X}{\mathrm{argmax}}p\left(X,\gamma ,\sigma^{2}|Y\right)=\underset{X}{\mathrm{argmax}}p\left(X|Y,\gamma ,\sigma^{2}\right)p(\gamma ,\sigma^{2}|Y)$$
where γ and $\sigma^{2}$ are the source and noise variances, respectively. As shown in the right term of Equation (5), the estimation was conducted in two phases, $p\left(X|Y,\gamma ,\sigma^{2}\right)$ and $p(\gamma ,\sigma^{2}|Y)$.

First, variances γ and $\sigma^{2}$ were obtained by maximizing $p(\gamma ,\sigma^{2}|Y)$ using the given measurement Y. In this study, because the variances were assumed to follow a uniform distribution as in a previous study [27], the right term $p(\gamma ,\sigma^{2}|Y)$ of Equation (5) was equivalent to $p(Y|\gamma ,\sigma^{2})$. Then, the solution $\hat{X}$ was derived with a maximum a posteriori (MAP) estimation of the left term $p\left(X|Y,\gamma ,\sigma^{2}\right)$ with the given measurement Y and the optimal variances from the right term $p(\gamma ,\sigma^{2}|Y)$.

Using Baye’s theorem, the posterior probability distribution $p\left(X|Y,\gamma ,\sigma^{2}\right)$ can be denoted as follows [28]:(6)$$p\left(X|Y,\gamma ,\sigma^{2}\right)=\frac{p(Y|X,\sigma^{2})p(X|\gamma )}{p(Y|\gamma ,\sigma^{2})}$$

Here, $p(Y|X,\sigma^{2})$ and p(X|γ) are the likelihood and prior probability, respectively. The denominator $p(Y|\gamma ,\sigma^{2})$ was the evidence used to estimate the variance of the source and noise.

Because noise is an independent random variable following a zero-mean complex Gaussian distribution, the likelihood $p(Y|X,\sigma^{2})$ is expressed as follows using the relation between measurement Y and noise $\sigma^{2}$ [28]:(7)$$\begin{array}{cc} p(Y|X,\sigma^{2})= & \overset{L}{\underset{l=1}{\prod }}\frac{1}{{\left(\pi \sigma^{2}\right)}^{M}}\exp\left(-\sigma^{-2}{\left|\left|y_{l}-Ax_{l}\right|\right|}^{2}\right)\\ & \;\;=\frac{1}{{\left(\pi \sigma^{2}\right)}^{ML}}\exp\left({-\sigma^{-2}{\left|\left|Y-AX\right|\right|}_{}^{2}}_{F}\right). \end{array}$$

Here, ${{\left|\cdot \right|}_{F}}_{}$ is a matrix Frobenius norm.

In MM-SBL, because unknown X is treated as a zero-mean Gaussian random variable with covariance matrix in the form of Γ=diag(γ), the prior is expressed as follows [28]:(8)$$p\left(X|\gamma \right)=\overset{L}{\underset{l=1}{\prod }}\frac{1}{\pi^{N}\det\left(\Gamma \right)}\exp\left(-x_{l}^{H}\Gamma x_{l}\right)=\frac{1}{{\left(\pi^{N}\det\left(\Gamma \right)\right)}^{L}}\exp\left[-\mathrm{tr}\left(X^{H}\Gamma X\right)\right]$$
where tr(·) is the trace of the matrix, ${\left(\cdot \right)}^{H}$ is the Hermitian transpose of matrix and det(·) is the determinant.

The evidence term is defined as the probability distribution of Y using γ and $\sigma^{2}$ in a linear system. Owing to the Gaussian likelihood and prior, the evidence term $p(Y|\gamma ,\sigma^{2})$ is also a Gaussian distribution and is expressed as follows [28]:(9)$$\begin{array}{cc} p(Y|\gamma ,\sigma^{2})= & \overset{L}{\underset{l=1}{\prod }}\frac{1}{\pi^{M}\det\left(\Sigma_{y}\right)}\exp\left(-y_{l}^{H}\Sigma_{y}^{-1}y_{l}\right)\\ & \;\;=\frac{1}{{\left[\pi^{M}\det\left(\Sigma_{y}\right)\right]}^{L}}\exp\left[-\mathrm{tr}\left(Y^{H}\Sigma_{y}^{-1}Y\right)\right] \end{array}$$
where $\Sigma_{y}=\sigma^{2}I_{M}+A\Gamma A^{H}$ is a covariance matrix of Y.

By substituting Equation (7) to Equation (9) into Equation (6), the posterior probability distribution $p(X|Y,\gamma ,\sigma^{2})$ can be expressed as follows [28]:(10)$$\begin{array}{cc} p(X|Y,\gamma ,\sigma^{2})= & \overset{L}{\underset{l=1}{\prod }}\frac{1}{\pi^{N}\det\left(\Sigma_{x}\right)}\exp\left[-{\left(x_{l}-\mu_{x_{l}}\right)}^{H}\Sigma_{x}^{-1}\left(x_{l}-\mu_{x_{l}}\right)\right]\\ & \;\;=\frac{1}{{\left[\pi^{N}\det\left(\Sigma_{x}\right)\right]}^{L}}\exp\left\{-\mathrm{tr}\left[{\left(X-\mu_{X}\right)}^{H}\Sigma_{x}^{-1}\left(X-\mu_{X}\right)\right]\right\} \end{array}$$
where the posterior mean $\mu_{X}=\Gamma A^{H}\Sigma_{y}^{-1}Y$, the posterior covariance $\Sigma_{x}={\left(\sigma^{-2}A^{H}A+\Gamma^{-1}\right)}^{-1}$, and $\Sigma_{y}^{-1}=\sigma^{-2}I_{M}-\sigma^{-2}A\Sigma_{x}A^{H}\sigma^{-2}$.

Next, to obtain a solution using MM-SBL, the variances γ and $\sigma^{2}$ that best describe the given measurement Y are estimated by the following Equation [28]:(11)$$\left(\hat{\gamma },{\hat{\sigma }}^{2}\right)=\underset{\gamma ,\sigma^{2}}{\mathrm{argmax}}p(Y|\gamma ,\sigma^{2})$$

To update the variances γ and $\sigma^{2}$, we applied type-II maximum likelihood and stochastic maximum likelihood, respectively, as in previous studies [27,28]. First, variance γ is estimated by maximizing the evidence of the log-likelihood form as follows:(12)$$\hat{\gamma }=\underset{\gamma }{\mathrm{argmax}}\log p(Y|\gamma ,\sigma^{2})\propto \underset{\gamma }{\mathrm{argmax}}\left[-\mathrm{tr}\left(Y^{H}\Sigma_{y}^{-1}Y\right)-L\log{\det\Sigma }_{y}\right]$$

By differentiating the objective function of Equation (12) by γ, we obtain the gamma that maximizes the objective function. The optimal gamma $\hat{\gamma }$ is obtained by repeating the update rule in the loop of the MM-SBL [28]:(13)$$\gamma_{n}^{new}=\frac{\gamma_{n}^{old}}{\sqrt{L}}{\left|\left|Y^{H}\Sigma_{y}^{-1}a_{n}\right|\right|}_{2}^{2}/\sqrt{a_{n}^{H}\Sigma_{y}^{-1}a_{n}}$$

$\gamma_{n}^{new}$ and $\gamma_{n}^{old}$ are the updated and present values of $\gamma_{n}$, respectively, and $a_{n}$ is the nth column vector of A.

To update the variance $\sigma^{2}$, we adopted the stochastic maximum likelihood [28,35] and ${\hat{\sigma }}^{2}$ was estimated by
(14)$${\left(\sigma^{2}\right)}^{new}=\frac{\mathrm{tr}\left[\left(I_{M}-A_{ℳ}{A^{+}}_{ℳ}\right)YY^{H}\right]}{L\left(M-K\right)}$$

ℳ ={n∈ℕ|K largest peaked in γnew} is the active set, A is the matrix that contains K active columns of matrix A, and A+ is the Moore–Penrose pseudo-inverse of matrix A**;** the number of active columns K is defined by the user, and any choice 0≤K≤M does not have a significant impact on performance [28]. In the MM-SBL, the solution X^ is computed with the average of μX along the measurements, and the shared common supports improve the robustness of the MM-SBL.

## 5. Experimental Results Using In-Situ Underwater Acoustic Data

In this section, we validate target identification and tracking performance using the MM-SBL with underwater in-situ data. We compare its performance with that of conventional target detection method.

### 5.1. Data Description

The target identification and tracking using the MM-SBL were applied to the underwater in-situ data gathered near the Korean Peninsula. As shown in Figure 4a, the experimental site is almost flat and shallow with a depth of 70 m. The measurement data were recorded for 66 min using a uniform horizontal line array (HLA) on the sea bottom, composed of 48 sensors. The experimental ship moved at a speed of approximately 1–1.5 m/s and passed through the sensor array for approximately 30 min. In this paper, because we used passive sonar systems developed for the defense system, information such as the sampling frequency, design frequency of the sensor array, distance of each sensor, and frequency component of the radiation signal, cannot be disclosed. Thus, all frequency analysis results were normalized by the maximum frequency of fs/2.

Figure 4b shows the conventional beamforming results using the strongest signal (i.e., beamforming at the frequency corresponding to the pilot signal), and it indicated the path of the experimental ship, which is confirmed by the global positioning system (GPS) of the experimental ship. During the data measuring, there were several fishing boats. Figure 4c shows the measuring results of broadband beamforming using all frequency components at intervals of 1 Hz, and displays the paths of the fishing boats with the experimental ship.

### 5.2. Experimental Results of Target Identification and Tracking

Figure 5 shows the FRAZ plots of the conventional and MM-SBL based target detection methods using in-situ underwater acoustic data from 15 min 20 s to 15 min 22 s, where the experimental ship was located relatively far from the sensor array. As shown in Figure 4c, the result of the broadband beamforming using all frequency components shows that there are several ships, including fishing boats (at approximately −60°, −33°, −22°, 30° and 65°) and the experimental ship (at approximately 9°). Between −30° and −20°, there are some fishing boats, including one with a stronger signal power than the experimental ship (see the red box in Figure 4c). For the convenience of comparison, the results from the conventional and MM-SBL based detection methods were normalized with the corresponding maximum amplitudes after frequency analysis and DOA estimation.

Figure 5a–c show the FRAZ plots obtained using the conventional detection method after applying a threshold of 0.05. As shown in Figure 5a–c, although the same threshold value was applied, the conventional detection method had different target detection results according to time (five targets in Figure 5a, six targets in Figure 5b, and four targets in Figure 5c). In addition, some false alarm detection occurred due to sidelobes (in particular, near 0.4, 0.5, 0.77, and 0.83 of the normalized frequency).

Figure 5d–f show the FRAZ plots obtained using MM-SBL. For target detection, we performed narrowband beamforming using common frequency components detected by frequency analysis using MM-SBL. For DOA estimation using the MM-SBL, we used 30 multiple measurements (30 multiple-time snapshots with 90% overlap). In Figure 5d, the MM-SBL detected six targets (four targets near −60°, 9°, 30° and 65° and two targets near −30°), which one additional target near −20° compared to the conventional target detection method at 15 min 20 s (see Figure 5a). The MM-SBL improves the detection results by finding common signal components with multiple measurements. In addition, the results of false alarm detection were significantly reduced compared to the conventional detection method (in particular, between 0.77 and 0.83 of the normalized frequency). Figure 5e,f show the detection results at 15 min 21 s and 15 min 22 s, respectively. Unlike the conventional detection method, the detection results using MM-SBL presented a constant number of six targets along time.

From the characteristics of the frequency components of each target in Figure 5, most tonal frequency components of the fishing boats (at approximately −60°, −30°, 30° and 65°) were distributed below 0.5 of the normalized frequency. On the other hand, most of the tonal frequency components of the experimental ship (at approximately 10°) were distributed above 0.5 of the normalized frequency (higher tonnage than fishing boats). Thus, we can identify the target according to the frequency distribution of the FRAZ plot results.

Figure 6 shows an enlarged view of the frequency component between 0.15 and 0.525 and the azimuth angle between 45° and 75° of the FRAZ plots in Figure 5. The target detection method using the MM-SBL has clearer results than the conventional target detection method. The target detection results have almost the same frequency components for the target near an azimuth of 65° along time. On the other hand, the conventional target detection method has ambiguous detection results owing to ambient noise, and the detected frequency components are inconvenient.

Figure 7 shows the DOA estimation and frequency analysis results of the MM-SBL (black dotted line) and the conventional method (blue line) using the underwater acoustic data at 15 min 21 s (see the red box in Figure 4c). In Figure 7, we did not apply the threshold and local maximum for the conventional detection method because the detection results obtained using the conventional method depend on the threshold value.

Figure 7a,b show the frequency analysis results of the MM-SBL and FFT at azimuth angles of −20° and 9°, respectively. Here, −20° and 9° represent the angles of the fishing boats and experimental ship, respectively. The conventional detection method enhances the signal of a specific azimuth through beamforming using the sensor array and performs frequency analysis on the signal using FFT. However, as mentioned in Section 3, target detection using MM-SBL finds the common frequency components commonly contained in the signal through frequency analysis using signals arriving at the sensor array (consisting of 48 sensors). After that, we estimated the azimuth represented by each common frequency component using MM-SBL based beamforming. For the frequency analysis using MM-SBL, we use 48 sensors and 30 multiple-time snapshots (i.e., 1440 multiple measurements). In Figure 7a,b, the frequency analysis results of the MM-SBL in Figure 7a,b show the frequency components indicating −20° and 9° among the selected common frequency components. As can be seen in Figure 7a,b, frequency analysis using FFT has difficulty specifying the frequency components of the target due to ambient noise. However, the MM-SBL reduces the noise around the meaningful signal through multiple measurements and has high resolution results; thus, the frequency components of the target existing in a specific azimuth can be clearly identified.

Figure 7c,d show the narrowband DOA estimation results of conventional beamforming (CBF) and MM-SBL at the normalized frequency components of 0.349 and 0.653, respectively. Here, the normalized frequency components 0.349 and 0.653 were identified as tonal frequency components belonging to the fishing boat and experimental ship, respectively. For DOA estimation, we used a sensor array consisting of 48 sensors. DOA estimation using the MM-SBL was applied to 30 multiple measurements (30 multiple-time snapshots). The MM-SBL has enhanced DOA estimation results over the conventional detection method by suppressing the noise signal and finding consistent target signal components using multiple measurement data.

Figure 7e shows a FRAZ plot of the proposed detection method using the MM-SBL at 15 min 21 s. ‘A’ and ‘B’ in Figure 7e mean frequency lines at steering angle −20° and 9°, respectively. ‘C’ and ‘D’ in Figure 7e indicate azimuth lines at the normalized frequency components of 0.349 and 0.653, respectively.

Figure 8 presents the ship tracks by integrating the target detection results along time (15 min, 17 min, and 23 min). In Figure 8, we repeatedly performed frequency analysis and DOA estimation using the underwater in-situ data at different times. We summarized the detection results at each time as the sum of the signal spectrum at a certain steering angle (i.e., the sum of the FRAZ plot in Figure 5 over the vertical axis(frequency)). In Figure 8, ‘A’ is the path of the experimental ship, which is confirmed by the GPS, and ‘B’ is the path of the fishing boat with the strongest signals; path ‘B’ intersected the path of the experimental ship between 15 and 23 min (see the red and yellow box in Figure 4c). As seen in the red box of Figure 4c, six targets exist (one experimental ship and five fishing boats) between 15 to 23 min, and most fishing boats are near −30°.

Figure 8a–c show the ship tracks at 15, 17, and 23 min using the conventional detection method. In Figure 8a–c, we did not apply the threshold and local maximum because the detection results using the conventional method depend on the threshold value. In Figure 8a–c, the conventional detection method unclearly shows target tracking results owing to ambient noise. As shown in Figure 8a,b, the conventional detection method cannot distinguish targets (fishing boats) at approximately −30°. In addition, for targets detected at an endfire, it was difficult to estimate the angle of the target accurately because the detection results were smeared. Figure 8c shows the detection results when the experimental ship (‘A’) and fishing boat (‘B’) were close. The conventional detection method cannot distinguish targets ‘A’ and ‘B’, and it detects targets near −60°, −30° and 30° unclearly.

We confirmed that the benefits of MM-SBL (high-resolution and noise reduction) enable precise and accurate detection of target signals, as shown in Figure 8. As previously mentioned, the MM-SBL provides high-resolution detection results by finding common components in multiple measurement data and effectively suppressing noise (approximately −30°). Figure 8d–f show the ship tracks using the MM-SBL. As seen in Figure 8d,e target detection using the MM-SBL can distinguish adjacent signals that the conventional detection method cannot distinguish (near −30°). In addition, the MM-SBL, which efficiently reduced noise, exhibited relatively clear detection results near the endfire (between 60° and 90°). In Figure 8f, target tracking results using the MM-SBL show a distinction between two close targets (‘A’ and ‘B’) clearly and precisely, and targets near −60° and −30° were also clearly detected. We confirmed that we could track the target’s path through these results over time.

## 6. Discussion

Here, we performed ‘identification’ with the presumption of frequency components of ship differing along ship types and displayed the detected frequency components according to azimuth angles via MM-SBL, which should be helpful to identifying the ship types with the corresponding locations. The simulation results using in-situ underwater acoustic data were compared to the traditional target detection method. The traditional target detection method, which reinforce SNR of the signal by forming a beam at a specific angle, detect frequency components of the target by analyzing the frequency of the beam signal where the target is located. The traditional method has low resolution and blur due to ambient noise around the target signal, making it difficult to distinguish targets. In this paper, frequency analysis was performed before beamforming and the tonal frequency component in underwater acoustic data was detected, unlike traditional target detection method, using the characteristic of the MM-SBL that can detect common components present in underwater acoustic signal data. Thereafter, we performed the MM-SBL based on narrowband beamforming to estimate the azimuth angle at which each detected frequency component is located and identified the targets. Recall that MM-SBL effectively detects and estimates common components of signal through multiple measurements. As demonstrated in previous studies [34], frequency analysis of sparse signals using MM-SBL has advantages in term of improved resolution and noise reduction. In this paper, we proposed a target identification method from a new perspective using MM-SBL, which has been previously validated, and examined the performance using in-situ underwater acoustic data. Additionally, we can track the path of the target by arranging the target detection results over time and can distinguish close targets through the high-resolution beamforming results of MM-SBL.

## 7. Conclusions

We proposed a target detection method using MM-SBL for the identification and tracking of marine objects. Unlike, the conventional detection method, which identifies the target through frequency analysis for a specific azimuth angle after beamforming, we selected common frequency components of the data by frequency analysis using MM-SBL. Subsequently, we performed the DOA estimation for each common frequency component using MM-SBL based beamforming. In the FRAZ plot, we confirmed the azimuth of the target signal for each common frequency component and verified the alignment of the unique frequency components of the target signal. Based on these results, we identified targets through DOA estimation for each frequency component. In addition, we could track the path of the target by using target detection results along time, which are derived from the sum of the signal spectrum at the azimuth angle. Using in-situ data gathered near the Korean peninsula, we confirmed that the target detection method using MM-SBL has high-resolution detection results compared to the conventional target detection method.

## Figures and Tables

**Figure 1 sensors-22-08511-f001:**
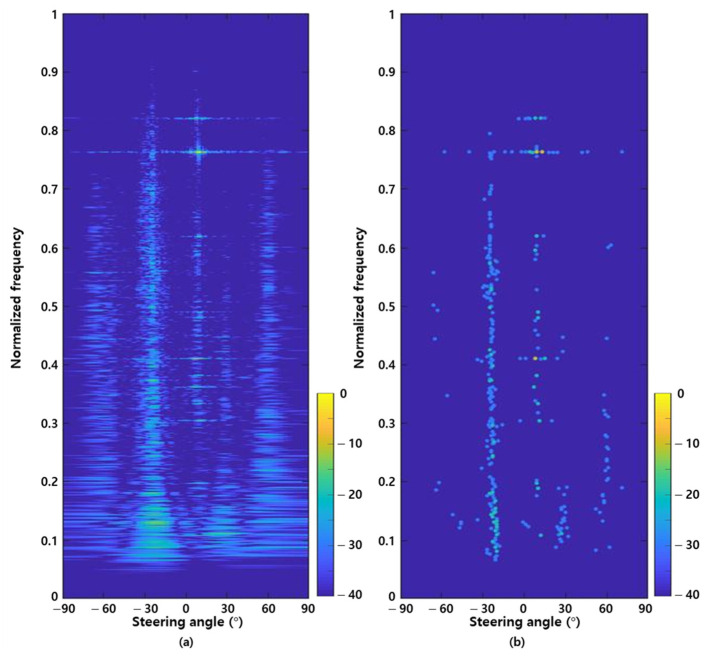
Frequency-azimuth (FRAZ) plots using conventional target detection method; (**a**) detection results without detection threshold and local maximum; (**b**) detection results with detection threshold and local maximum.

**Figure 2 sensors-22-08511-f002:**
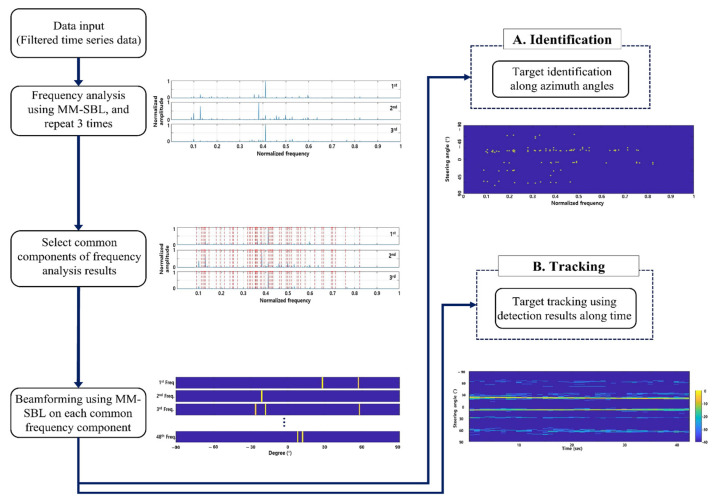
Flowchart of target identification and target tracking using MM-SBL.

**Figure 3 sensors-22-08511-f003:**
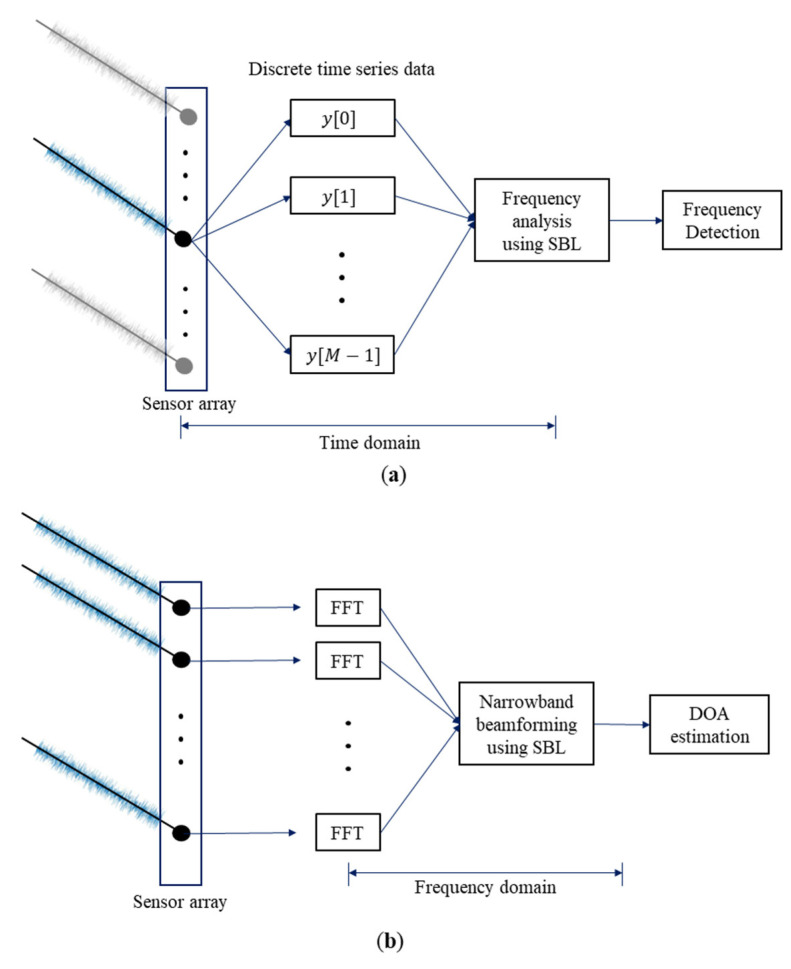
Process of (**a**) the frequency analysis using single measurement; (**b**) the DOA estimation using single measurement.

**Figure 4 sensors-22-08511-f004:**
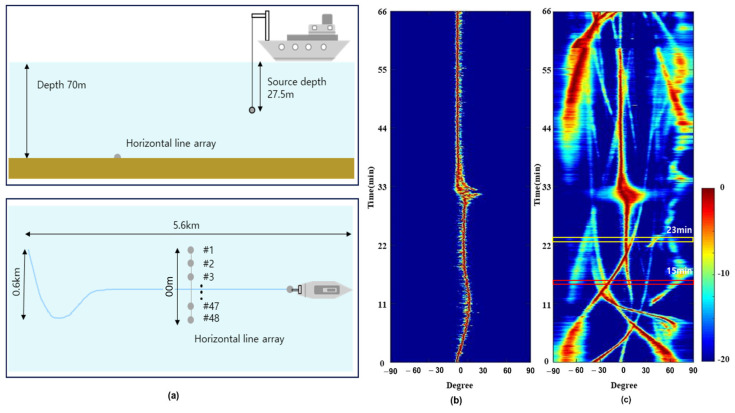
(**a**) Experimental setup of horizontal line array and source experimental ship; (**b**) The path of experimental ship from conventional beamforming method using pilot frequency component; (**c**) The broadband beamforming results using all frequency components.

**Figure 5 sensors-22-08511-f005:**
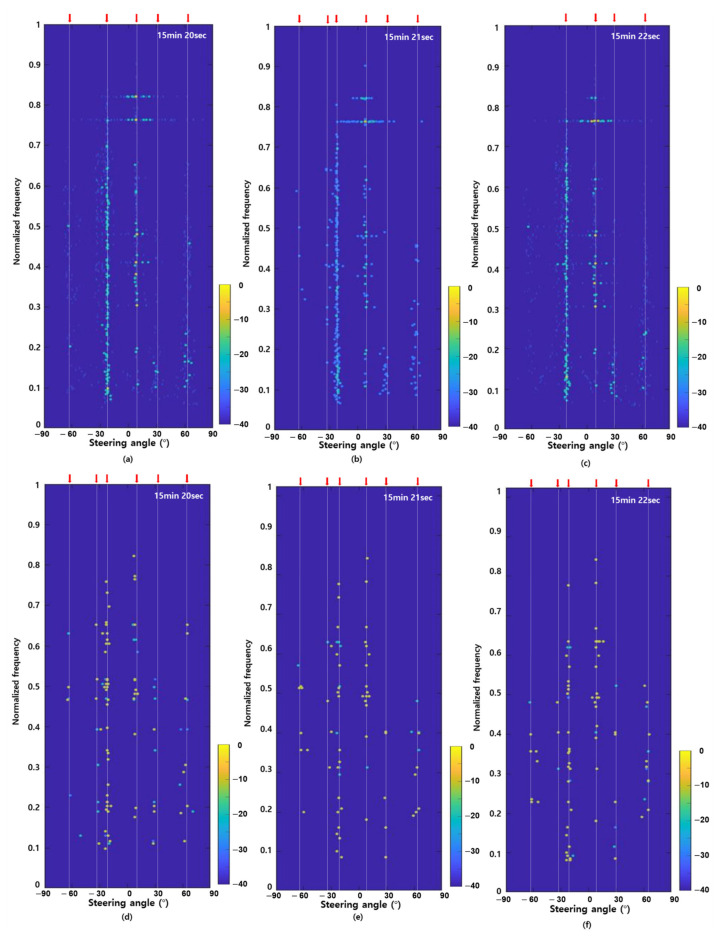
Frequency-azimuth plots using (**a**–**c**) conventional detection method with detection threshold; (**d**–**f**) proposed detection method using the MM-SBL at the 15 min 20 s to 22 s.

**Figure 6 sensors-22-08511-f006:**
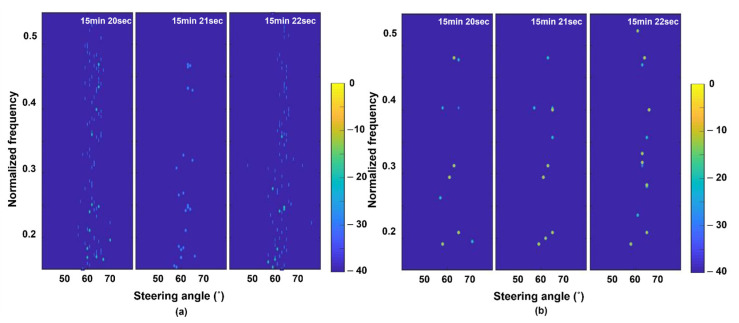
Refined frequency-azimuth plots of (**a**) conventional detection method; (**b**) proposed detection method using the MM-SBL.

**Figure 7 sensors-22-08511-f007:**
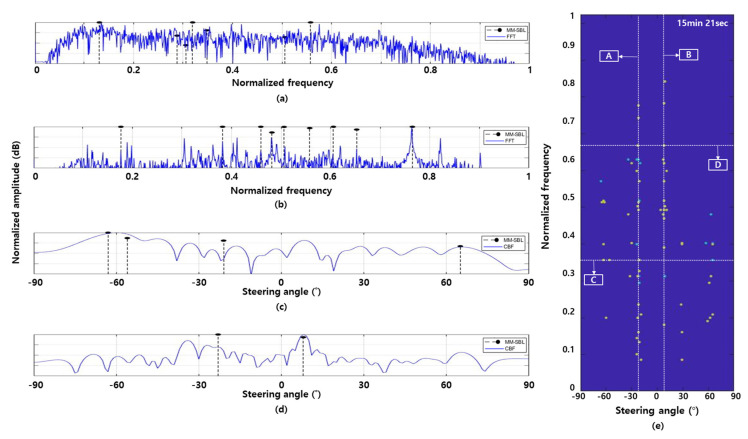
Frequency analysis results at azimuth of (**a**) −20° (fishing boat); (**b**) 9° (experimental ship); and the DOA estimation results at normalized frequency component of (**c**) 0.349 (fishing boat); (**d**) 0.653 (experimental ship); and (**e**) FRAZ plot of the proposed detection method using the MM-SBL at 15 min 21 s. A and B in (**e**) mean frequency lines at steering angle −20° and 9°, respectively. C and D in (**e**) mean azimuth lines at normalized frequency component of 0.349 and 0.653, respectively.

**Figure 8 sensors-22-08511-f008:**
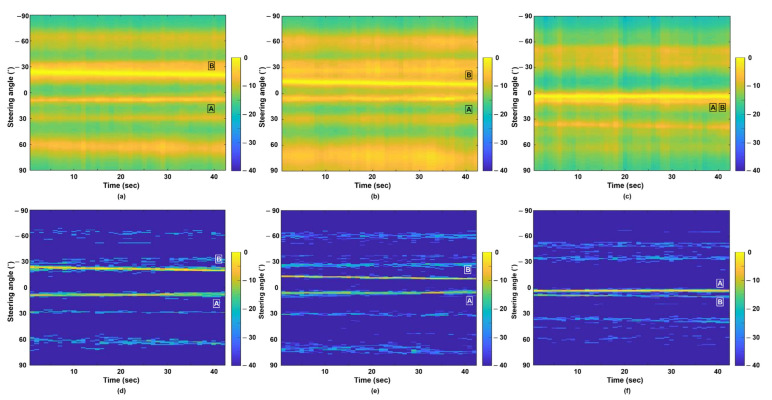
Detection results along time using (**a**–**c**) conventional detection method; (**d**–**f**) proposed detection method using the MM-SBL; at (**a**,**d**): 15 min; (**b**,**e**) 17 min; (**c**,**f**) 23 min.

## Data Availability

Not applicable.

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
