# Peer review of "Passive Sonar Target Identification Using Multiple-Measurement Sparse Bayesian Learning"

_sensors, 2022, doi:10.3390/s22218511_

Round 1

Reviewer 1 Report

The authors' MM-SBL approach is effective for measuring numerous targets and determining the direction of arrival of each detected frequency component based on beamforming MM-SBL. Only one recommendation can be added to the fourth section to strengthen the theory, and it should only be used as a guide.

Author Response

We thank the reviewer for careful and constructive comments. We have responded to the reviewer’s comments as following attachment.

Reviewer 2 Report

In this paper, the author used the method based on multiple-measurement sparse Bayesian learning (MM-SBL) to process the data obtained from the passive sonar system, so as to achieve target identification and tracking. This method analyzes the frequency by detecting the common frequency components of the signals obtained from different sensors in the array, and finally realizes the DOA estimation of the target. The author compares the results of this method with the conventional target detection method (ED method) to prove that it has better detection performance. The method described in this paper is logically complete and has certain application prospects. After reading this paper, I put forward the following suggestions for improvement:

1. I noticed that the author published a paper in the same journal last year [1], and compared with this paper [2], there seems to be a high overlap in the theoretical part. The former's frequency analysis based on multiple-measurement sparse Bayesian learning described is also the core content of the method described in this paper. Of course, the purpose of these two papers is different. The author applied this method to target identification and tracking, which verified the effectiveness of the method. But I still want to know, compared with the former, is there any significant improvement in the method of this paper?

[1] Frequency Analysis of Acoustic Data Using Multiple-Measurement Sparse Bayesian Learning

[2] Passive Sonar Target Identification using Multiple-Measurement Sparse Bayesian Learning

2. The author's use of abbreviations is not formal. For example, the author writes on lines 363-364: "Figures 7c and Figure 5d show the narrowband DOA estimation results of CBF and MM-SBL at the normalized frequency components of 0.349 and 0.652, respectively. "So what does CBF represent here? I guess CBF is the abbreviation of conventional beamforming, but the author should write the full name and corresponding abbreviation when using it for the first time (in line 275 of the manuscript). There are also similar problems with other abbreviations in the text, please check and revise it.

3. There are some presentation errors in the manuscript, such as "Figures 7c and Figure 5d" in line 363. Did the author mean to say "Figures 7c and Figure 7d"? Another example is "frequency components of 0.349 and 0.652" in line 364. Is the data here 0.652 or 0.653? The same data in line 337, line 339 and line 375 is 0.653, while in line 365 and here is 0.652. Please keep consistent.

4. There are some writing errors in the manuscript. For example, the "Iidentification" in the "Experimental Results of Target Iidentification and Tracking" in line 285 is a wrong spelling, and the "haves" in line 10 is also a writing error. Please carefully check whether there are other similar errors in the manuscript.

Author Response

(The authors gave the same response as above.)

Reviewer 3 Report

The paper is devoted to a problem with a long history of study. The authors provide necessary references and prove the novelty of their research. The paper is well-written. Methods are described so that the reader can understand them. Experimental results are clearly presented. However there are a couple of remarks regarding this paper.

1. The term ‘identification’ is used an unusual way. Identification usually means a procedure to decide which class a target belongs to. But in context of this paper identification is a mix of advanced detection and tracking. I.e. each primary detected target is checked, confirmed and then tracked. This process might be call as identification too, but this should be clearly discussed in the text.

2. A human can easier interpret for a broadband plot on Fig. 1a rather than a set of dots on picked frequencies in Fig. 1b. Maybe a set of dots better suits further processing. But it might happen so that splitting a broadband noise on a set of tones is not the optimal way of processing.

Author Response

(The authors gave the same response as above.)

Round 2

Reviewer 2 Report

The author has modified as required.